# Drug Development from Natural Products Based on the Pathogenic Mechanism of Asthma

**DOI:** 10.3390/ijms241512469

**Published:** 2023-08-05

**Authors:** Min-Hee Kim, Chun-Sik Bae, So-Hyeon Bok, Hyo-Seung Choi, Taeho Ahn, Seung-Sik Cho, Dae-Hun Park

**Affiliations:** 1Department of Forestry and Landscape Architecture, Dongshin University, Naju 58245, Republic of Korea; minhee3947@dsu.ac.kr; 2College of Veterinary Medicine, Chonnam National University, Gwangju 61186, Republic of Korea; csbae210@jnu.ac.kr (C.-S.B.); thahn@jnu.ac.kr (T.A.); 3College of Oriental Medicine, Dongshin University, Naju 58245, Republic of Korea; bok_23@naver.com; 4Department of Digital Contents, Dongshin University, Naju 58245, Republic of Korea; design@dsu.ac.kr; 5Department of Pharmacy, College of Pharmacy, Mokpo National University, Muan 58554, Republic of Korea; 6Biomedicine, Health & Life Convergence Sciences, BK21 Four, College of Pharmacy, Mokpo National University, Muan 58554, Republic of Korea

**Keywords:** asthma, drug development, natural products, adverse effects

## Abstract

Asthma is a chronic inflammatory disease of the pulmonary system associated with many wheeze-to-sleep apnea complications that may lead to death. In 2019, approximately 262 million patients suffered from asthma, and 455 thousand died from the disease worldwide. It is a more severe health problem in children and older adults, and as the aging of society intensifies, the problem will continue to worsen. Asthma inducers can be classified as indoor and outdoor allergens and can cause asthma due to their repeated invasion. There are several theories about asthma occurrence, such as the imbalance between Th1 and Th2, inflammation in the pulmonary system, and the abnormal apoptosis/cell proliferation of cells related to asthma. Although there are many medications for asthma, as it is an incurable disease, the purpose of the drugs is only to suppress the symptoms. The current drugs can be divided into relievers and controllers; however, as they have many adverse effects, such as immune suppression, growth retardation, promotion of cataracts, hyperactivity, and convulsions, developing new asthma drugs is necessary. Although natural products can have adverse effects, the development of asthma drugs from natural products may be beneficial, as some have anti-asthmatic effects such as immune modulation, anti-inflammation, and/or apoptosis modulation.

## 1. Introduction

### 1.1. Definition

Asthma is an incurable chronic inflammatory disease with a hyper-responsive reaction in the pulmonary system [1]. In particular, the Global Initiative for Asthma defines asthma as “a heterogeneous disease, usually characterized by chronic airway inflammation”. It is defined by a history of respiratory symptoms, such as wheezing, shortness of breath, chest tightness, and cough. Various signs are observed according to the suffering time and intensity, as they are caused by variable expiratory airflow limitation [2]. Many patients simultaneously experience asthma, chronic obstructive pulmonary disease (COPD), and obstructive sleep apnea. This coexistence can lead to patient death; therefore, an advanced therapeutic protocol should be developed [3].

### 1.2. Epidemiology

Worldwide, 262 million patients had asthma, and 455,000 people died owing to the disease in 2019. Asthma is more severe in children and older adults than in young people [4]. In 2019, the average life expectancy (LE) of males and females with asthma was 70.9 and 75.9 years, respectively. In addition, the prospect is that asthma patients will increase rapidly [5]. Although asthma is globally prevalent, its morbidity and mortality rates are much higher in low- and middle-income countries than in developed countries. Considering the differences among continents and countries in Africa, it is most common in children and adolescents. Although many efforts have been made to control asthma in Latin America, it is not easy to effectively prevent it because of barriers such as insufficient public health insurance, social factors, economic factors, and political factors. In Argentina, the prevalence of asthma in older adults was 21.4% in 2019. In Brazil, the rates of severe asthma and mortality have decreased. In the Eastern Mediterranean Region, the prevalence of severe asthma is high; although some countries have a stable healthcare system, others do not. In Europe, the prevalence and mortality rates in some countries are much lower than in others, and this difference depends on each country’s situation. In Southeast Asia, there has been no considerable change in the occurrence of asthma. In the Western Pacific region, diverse aspects are observed based on culture, economy, politics, and geography [2]. The prevalence of asthma increases the burden of hospitalization in children, especially those who are 5 years old or younger [6].

In this review, we describe the pathogenesis of asthma, the problems with current asthma drugs, and the development of natural drugs and their advantages for asthma treatment.

## 2. Pathogenesis of Asthma

### 2.1. Asthma Inducers

The major causes of asthma are complex factors that contribute to confuse the immune system in bio-organisms. There are many asthma inducers, such as allergens, pollutants, tobacco smoking, cold temperature, and genetic background [7]. Patients with non-allergic asthma (NA) may comprise 10–33% of the asthma cases. Intrinsic factors, such as genetic background, are the cause of NA; however, in the related study, only allergic asthma was described [8]. Although many allergens exist, they can be classified into two categories: indoor and outdoor. Indoor allergens include house dust mites (Der p 1 and Der p 2, from *Dermatophagoides pteronyssinus*), cockroaches (Bla g 1, Bla g 2, and Per a 1), pet dander (Fel d 1 from cat and Can f 1 from dog), and outdoor allergens include mice (Mus m 1), pollen (Phl p1 and Phl p5, from Timothy and Amb a 1, 2, 3, 5, and 6, from ragweed), peanut proteins (Ara h 1, 2, and 3), and mold (*Alternaria*, *Cladosporium*, and *Epicoccum*) [9,10]. Recently, the relationship between asthma occurrence and air pollutants, such as traffic-related air pollution, sulfur dioxide, carbon monoxide, heavy metals, and polycyclic aromatic hydrocarbons, has also been reported [11]. The prevalence and incidence of asthma in smokers are higher than in non-smokers, but the more severe aspect is non-smoker-related second-hand smoking, which is especially dangerous for children [12]. Cold weather can induce asthma symptoms that are more severe in patients with uncontrolled asthma than in those with well-controlled asthma [13]. Further, more than 100 genes have been shown to cause asthma, such as the BsmI (rs1544410) and ApaI (rs7975232) polymorphisms of the vitamin D receptor gene, the *rs20541* and *rs1800925* polymorphisms of IL13, and the *rs4950928*, *rs10399931*, and *rs8883125* polymorphisms of chitinase 3-like 1 [14,15,16].

### 2.2. Histopathological Changes in Asthma Occurrence

Patients with asthma have problems ranging from wheezing to death due to apnea, and these problems are caused by obstacles to the pulmonary system. Airway remodeling, a representative change in the respiratory system, occurs from histopathological deformations such as respiratory epithelial cell hyperplasia, mucous hypersecretion by goblet cell hyperplasia, metaplasia, submucosal gland hypertrophy, inflammatory cell infiltration of eosinophils and neutrophils, and airway smooth muscle contraction [17,18,19,20].

### 2.3. Mechanisms of Asthma Pathogenesis

#### 2.3.1. Imbalance of Th1, Th17, and Th2

The immune system is important in maintaining homeostasis in organisms, and its function can be classified into two categories: innate and acquired immunity [21]. In particular, acquired immunity is important in relation to a pathogen’s reinvasion, as it has a memory of the pathogen’s characteristics and can rapidly eliminate the reinvading pathogens. Although it is important to maintain the balance of helper T cell subfamilies such as Th1, Th2, and Th17 cells in bio-organisms, in diseases such as asthma, atopic dermatitis, and viral infection, the upregulation of Th2 cells is important and results in an imbalance of helper T cell subfamilies, leading to an imbalance of Th1, Th17, and Th2 cells [22]. In particular, Th17 cell abnormalities can induce neutrophilic asthma [20].

Regulatory T (Treg) cells are very important modulators of the balance between Th1 and Th2 cells, as they play a critical role in maintaining immune tolerance to allergens; however, when asthma occurs, the level of Treg cells changes (Figure 1) [23]. Notably, Th2 cell-related cytokines such as IL-4, IL-5, and IL-13 and Th17 cell-related cytokines such as IL-6 and TNF-α increase in patients with asthma [24]. IL-4 stimulates the activation of GATA-3, which is a Th2 cell transcription factor, through a positive feedback loop, leading to the secretion of IgE by activating B cells, inducing eosinophilia, and stimulating airway hyperreactivity [25]. IL-5 activates B cells and increases the eosinophil population [26]. IL-13 causes goblet cell metaplasia, bronchial hyperreactivity, and eosinophil extravasation by stimulating the expression of intercellular adhesion molecule 1 (ICAM-1) and vascular cell adhesion molecule 1 (VCAM-1) [27]. IL-6 stimulates inflammation, promotes IL-4 production, and suppresses Th1 and Th17 cell differentiation, whereas TGF-β stimulates Th17 cell differentiation [22,28]. TNF-α is related to airway hyperresponsiveness [29]. In contrast to the increments of Th2 cell- and Th17 cell-related cytokines, the level of Th1 cell-related cytokines, such as IFN-γ and IL-12, decreases in patients with asthma [30]. IFN-γ has incompatible functions; it acts as a stimulator and activates Th1 cell transcription factor (T-bet), and as a suppressor, it activates Th2 cell transcription factor (GATA-3), preventing eosinophilia and inhibiting asthmatic changes such as airway hyperreactivity, inflammation, and mucous hypersecretion in the respiratory system [31,32]. IL-12 is related to Th1 cell differentiation and the suppression of Th2 cell propagation [33].

#### 2.3.2. Inflammation in the Pulmonary System

Asthma is a heterogeneous inflammatory disease, and various inducers cause the infiltration of immune cells, such as eosinophils and neutrophils, which release reactive oxygen species (ROS) to eliminate them. After the elimination of foreign bodies, the redox imbalance maintains ROS overproduction, inducing oxidative stress in the respiratory system. This stress, including inflammation, damages the respiratory system and induces hyperresponsiveness, airway remodeling, and mucus hypersecretion (Figure 2) [34].

According to the development time, respiratory inflammation-related asthma can be classified into two categories, i.e., acute and chronic, with acute asthma presenting an early a late phase (Table 1) [35]. The early phase of the acute inflammation stage occurs within several minutes of allergen exposure due to the cross-linking of allergens with IgE, which binds to immune cells such as mast cells and basophils. The cross-linked allergens can then release immune mediators such as cytokines and chemokines to affect the functions of several organs, including vasodilation, vascular permeability, bronchoconstriction, and mucus hypersecretion. The late phase occurs within 2–6 h after allergen invasion; the peak occurs 6–9 h later, and several symptoms can be observed, such as wheezing, shortness of breath, and cough. These symptoms are caused by an increase in Th2 cell-related cytokines, such as IL-4, IL-5, and IL-13, via Th2 cell activation and an increase in white blood cell populations, such as eosinophils, basophils, and other leukocytes. The chronic inflammation stage of the respiratory system is caused by repeated allergen contact, which can change organ function via a continuous alteration of the extracellular matrix and constitutive cells, ultimately inducing asthma.

Toll-like receptors (TLRs) play an important role in innate and acquired immunity, and TLR cascades in the respiratory system are closely associated with asthma and COPD [36]. The commensal microbiota in the gastrointestinal tract controls innate and acquired immune responses and is related to the immune reaction at extraintestinal sites, such as in the respiratory system. However, the immune response of the commensal microbiota in the respiratory system is not caused by TLRs on pulmonary cells but is related to bacterial Nod-like receptors (NLRs) in the digestive system [37]. This indicates that the NLR on the microbiota binds to NLR ligands and induces alveolar macrophages to release oxygen species to kill invading bacteria.

#### 2.3.3. Apoptosis/Cell Proliferation of Respiratory Epithelial Cells

The epithelial cell layer is the most important barrier against foreign bodies and invading bio-organisms, and foreign bodies, including asthma inducers, can damage this area through inflammation [38]. Owing to their role as a barrier, the repeated damage and recovery of pulmonary epithelial cells results in their apoptosis, and new cells emerge to replace the functions of dead cells [39]. However, reports regarding the relationship between apoptosis and asthma are contradictory. Some studies showed that epithelial cell apoptosis is a representative change related to asthma occurrence [40,41]. However, other researchers found that the population of apoptotic cells significantly increases in patients with steroid-untreated asthma [42,43]. These contradictory results might be depend on the epithelial cell stage in relation to the elimination of damaged/dying cells, which is induced by various stimulators, as the neighboring epithelial cells of damaged/dying cells may engulf the apoptotic cells [44].

Apoptosis is a homeostatic process that is related to normal cell turnover, immune system development and function, hormone release, and embryonic development [45]. Its mechanism is classified into three types, i.e., intrinsic (mitochondrial), extrinsic (death receptor), and related to perforin/5ranzyme (Figure 3) [46], which are connected to the execution pathway. The intrinsic pathway is not related to the receptor and is initiated by increased mitochondrial permeability; its initiator is caspase-8 [47], and the Bcl-2 family of proteins, which controls mitochondrial permeability, is strongly involved in this pathway [48]. The Bcl-2 family of proteins can be divided into two categories: anti-apoptotic proteins, including Bcl-2, Bcl-x, Bcl-XL, Bcl-XS, Bcl-w, and BAG, and pro-apoptotic proteins, including Bcl-10, Bax, Bak, Bid, Bad, Bim, Bik, and Blk [45].

The extrinsic pathway is initiated by the activation of apoptosis-related transmembrane receptors, including FasL/FasR, TNF-α/TNFR1, Apo3L/DR3, Apo2L/DR4, Apo2L/DR5, and its initiator, caspase-9 [49,50,51,52,53]. The perforin/granzyme pathway is deeply involved in the release of perforin, which creates pores in the membrane using granzyme A or B [54,55]. The execution pathway is the final step of apoptosis, in all three above-mentioned pathways, i.e., intrinsic, extrinsic, and perforin/granzyme pathways. The execution pathway completes the representative and various morphological and biochemical changes of apoptosis, such as protein/DNA degradation, DNA fragmentation, chromatin condensation, apoptotic body formation, and apoptotic cell engulfing [45]. The initiator of the execution pathway is caspase-3, and there are several executioner caspases, such as caspase-6, caspase-7, and caspase-3 [56].

The apoptotic pathway is summarized as follows. The apoptotic pathway consists of two steps; the first step can be divided into three categories: intrinsic, extrinsic, and perforin/granzyme pathways. The intrinsic pathway is not related to transmembrane receptors but is strongly related to mitochondrial changes, and the initiator is caspase-3. The extrinsic pathway requires apoptosis-related receptors and ligand binding, and the initiator is caspase-8. The perforin/granzyme pathway is activated by perforin and granzymes A/B activation. The second step is the execution pathway, in which representative changes related to apoptosis are observed [45,46,47,48,49,50,51,52,53,54,55,56].

Although the relationship between apoptosis and proliferation is controversial, it is strongly related to cellular homeostasis, as it can block cellular proliferation and promote apoptotic events. The inhibition of pulmonary epithelial cell proliferation in patients with asthma might be similar to that observed in steroid-treated patients, in which the number of apoptotic cells increases [42,43]. The NF-κB/COX-2 pathway is related to both cell proliferation and inflammation, and NF-κB, as a transcription factor, induces the expression of COX-2. Finally, this pathway suppresses apoptosis when the number of tumor cells increases [57] and increases inflammation in patients with asthma [58]. Based on these studies, the search for inhibitors of the NF-κB/COX-2 pathway is one of the strategies to develop asthma drug candidates.

The pathogenesis of asthma can be attributed to various inducers and can be established by an imbalance between Th1 and Th2 cells, inflammation in the pulmonary system, and anti-apoptosis/cell proliferation of respiratory epithelial cells.

## 3. Asthma Medications

### 3.1. Asthma Drug Classification

Asthma drugs are classified as relievers (bronchodilators) or controllers. The relievers include anticholinergics (atrovent and tiotropium bromide), β-adrenergic drugs (salmeterol and formoterol), and methylxanthines (theophylline and aminophylline). These molecules are categorized as anti-inflammatory drugs or immunomodulators. Anti-inflammatory drugs include corticosteroids (dexamethasone, fluticasone, budesonide, mometasone, beclomethasone, and ciclesonide), leukotriene modifiers (montelukast, zafirlukast, and zileuton), and mast cell stabilizers (disodium cromoglycate, cromolyn, nedocromil, olopatadine, and ketotifen). Immune modulators can be classified into two groups: immunosuppressors (methotrexate and cyclosporin A) and immunomodulators, including immune potentiators (glucocorticoids, 1,25-dihydroxy vitamin D3, and TLR 2/4/9 ligands) [59]. The strategy of asthma treatment includes combining the above two categories of drugs, i.e., relievers and controllers (Table 2) [60]. Acetylcholine, a neurotransmitter in the parasympathetic nervous system, activates M3 muscarinic acetylcholine receptors that induce bronchoconstriction, and anticholinergics can inhibit the constriction of small airways [61]. β_2_-Adrenergic receptors abundantly exist on the airway smooth muscle (ASM), and the inspiratory–expiratory cycle in the pulmonary system is controlled by the binding of β-adrenergic compounds to β_2_-adrenergic receptors. β_2_-Adrenergic drugs are related to ASM relaxation [62]. The use of methylxanthines has gradually decreased because of their side effects. However, some methylxanthines, such as theophylline, are recommended as anti-asthmatic drugs because they have bronchodilatory and anti-inflammatory effects [63,64]. Inflammation begins with cell membrane destruction and ends with the production of leukotrienes or prostaglandins. Corticosteroids block the conversion of phospholipase A_2_ from the plasma membrane to arachidonic acid and can completely prevent inflammation. Inhalable corticosteroids are mainly used as anti-asthmatic agents [65,66]. Leukotrienes (LTs), which can be classified as cysteinyl-TLs (LTC_4_, LTD_4_, and LTE_4_) and LTB_4_, are pivotal for the occurrence of asthma. Cysteinyl-TLs induce airway remodeling and asthma exacerbation via inflammation [67,68]. The allergic response by mast cells is initiated by the interaction of the allergen with the IgE–FcεRI complex and maintained by chemical mediators such as histamine released by mast cells [69]. Mast cell stabilizers inhibit their activation and decrease asthma exacerbation [70].

### 3.2. Adverse Effects of the Current Drugs

Xie et al. reported that, in the USA from 2000 to 2016, 12,640 of 698,501 patients with asthma from 0 to 20 years of age (1.7%) experienced adverse effects of asthma drugs, and 0.83% of the adverse events were related to corticosteroids [71]. In particular, in patients aged 0–4 years, the incidence rate of asthma drugs’ adverse effects significantly increased from 0.2% to 19.3%. In the early stages of bronchodilator use, excessive doses of compounds such as isoprenaline cause toxicity in users [72]. The side effects of anticholinergics include a dry mouth, constipation, cough, headaches, and nausea; those of β_2_-adrenergic receptor agonists include trembling, nervous tension, headaches, muscle cramps, and heart attack; theophylline, a methylxanthine, causes nausea/vomiting, diarrhea, palpitation, tachycardia, arrhythmia, headaches, and insomnia [73]. The adverse effects of corticosteroids include cataract/glaucoma in the eye, hypertension/hyperlipidemia in the cardiovascular system, peptic ulcer/pancreatitis in the gastrointestinal tract, myopathy/osteoporosis in the skeletal muscle, dermal atrophy in the skin, immunological suppression, and growth retardation, especially in children [74,75]. The adverse effects of leukotriene modifiers are more common in children than in adults and are classified into psychiatric and non-psychiatric types. The psychiatric effects include hyperactivity, excessive sleepiness, nyctophobia, nervousness, agitation, hallucinations, and sleep disorders. The nonpsychiatric effects include abdominal pain, rash, aphthous ulcers, increased appetite, headache, and convulsions [76]. Mast cell stabilizers have several adverse effects, such as throat irritation, cough, anaphylaxis, and headache [77,78]. Immune modulators can help control asthma symptoms including nausea, increased serum aminotransferase, diarrhea, and hair problems [79].

The current drugs for asthma treatment are used only for preventing/relieving asthma symptoms, not for curing the disease, and they have many adverse effects; therefore, the development of new drugs is necessary.

**Table 2 ijms-24-12469-t002:** The adverse effects of asthma drugs.

Classification		Drugs	Adverse Effects	Refs.
Relievers (bronchodilators)	Anticholinergics	Atrovent, Tiotropium bromide, etc.	Dry mouth, constipation, cough, headache, nausea, etc.	[73]
	β-adrenergic drugs	Salmeterol, Formoterol, etc.	Trembling, nervous tension, headaches, muscle cramps, heart attack, etc.	
	Methylxanthines	Theophylline, Aminophylline, etc.	Nausea/vomiting, diarrhea, palpitation, tachycardia, arrythmia, headaches, insomnia, etc.	
Controllers	Anti-inflammatory drugs	Corticosteroids (Dexamethasone, Fluticasone, etc.)	Cataract/glaucoma, hypertension/hyperlipidemia, peptic ulcer/pancreatitis, myopathy/osteoporosis, dermal atrophy, immunological suppression, growth retardation, etc.	[74,75]
		Leukotrienes modifiers (Montelukast, Zafirlukast, etc.)	Hyperactivity, excessive sleepiness, nyctophobia, nervousness, agitation, hallucination, sleep disorder, abdominal pain, rash, aphthous ulcer, appetite increase, headache, convulsion, etc.	[76]
		Mast cell stabilizers (Cromolyn, Nedocromil, etc.)	Throat irritation, cough, anaphylaxis, headache, etc.	[77,78]
	Immunomodulators	Glucocorticoids, 1,25-dihydroxy vitamin D3, etc.	Nausea, increment of serum aminotransferase, diarrhea, hair problem, etc.	[79]

## 4. Natural Drugs for Asthma Treatment

As the current asthma drugs have many adverse effects, natural drug development might be an advisable method. In this section, studies on natural asthma drug development are classified according to the mechanism of asthma occurrence (Table 3).

### 4.1. Immune Modulators

Chrysin, which is 5,7-dihydroxyflavone and originates from propolis and passion flowers, controls asthmatic changes in a murine model by decreasing the levels of IgE, IL-4, and IL-13 [80]. 1′-Acetoxychavicol acetate isolated from *Alpinia galanga*, which is used as a traditional drug for gastric ulceritis, inhibits the release of Th2 cell-related cytokines, such as IL-4, IL-5, and IL-13, and Th1 cell-related cytokines, such as IL-12 and IFN-γ [81]. *Alginate oligosaccharide*, fermented by *Bacillus subtilis* KCTC 11782BP, inhibits the release of IL-5 and IL-13 [82]. *Allium cepa* L., which has long been used as an anti-inflammatory drug, downregulates Blomia tropicalis-induced Th2 cell-related cytokines, such as IL-4 and IL-13 [83]. Allium hookeri, which is used as a culinary material, significantly inhibits the release of Th2 cell-related cytokines, such as IL-4, IL-5, and IL-13 [84]. *Anoectochilus formosanus* HAYATA, a traditional drug, suppressed IgE, IL-4, and IL-5 levels in an asthma model [85]. Caenorhabditis elegans extracts decreased IgE, IL-5, and IL-13 and increased IFN-γ levels [86]. *Camellia japonica* oil, which is used as a hair cosmetic, controls ovalbumin-induced asthma via the GATA-3 and IL-4 pathways, and the active compound is oleic acid [87]. *Citrus tachibana*, a fruit, restored the balance of Th1/Th2 via the downregulation of Th2 cell-related cytokines such as IL-4 and IL-5 and the upregulation of Th1 cell-related cytokines such as IL-12 and IFN-γ, the imbalance of which was induced by ovalbumin [88]. Curcumin from *Curcuma longa*, which has long been used as an anti-inflammatory drug, significantly decreased asthmatic changes in the pulmonary system by downregulating the levels of GATA-3, a Th2 cell transcription factor [89]. Erythronium japonicum, a culinary material, suppressed Th1-related cytokines, such as IL-12p35 and IFN-γ, and Th2-related cytokines, such as IL-4, IL-5, and IL-13 [90]. Korean red ginseng exerted anti-asthmatic effects by modulating IL-12, IL-4, and IL-6 [91], and *Mycoleptodonoides aitchisonii*, a culinary material, significantly downregulated the levels of IL-4, IL-5, and IL-13 [92]. *Opuntia humifusa*, a culinary ingredient, suppressed ovalbumin-induced asthmatic changes by regulating Th1-/Th2-/Th17-related cytokines [93].

### 4.2. Anti-Inflammatory Effectors

Chrysin exerts an immunomodulatory and anti-inflammatory effects by downregulating Th2 cell-related cytokines and blocking inflammatory cell infiltration in the pulmonary system, respectively [80]. Three methoxy-catalpoides from *Psueolysimachion rotundum* var. subintegrum suppressed inflammation by decreasing the expression of COX-2 and iNOS and downregulating the levels of proinflammatory cytokines such as IL-1β, IL-6, and TNF-α [94]. *Aster yomensa* (Kitam). Honda, a culinary material, ameliorated inflammation through the downregulation of NO and IL-1β, which are related to TLR4 and NF-κB actions [95]. *Codonopsis laceolata*, which is used as a drug and as culinary material, suppressed asthma severity through NF-κB/COX-2 and PEG_2_ pathways [96]. *Echinodorus scaber rataj* controls inflammation by downregulating inflammatory cell migration [97]. Korean red ginseng suppressed asthma occurrence through the dual effects of immune modulation via IL-4 downregulation and inflammation inhibition through the NF-κB/COX-2 and PEG_2_ pathways [91]. *Pericampylus glaucus* inhibited inflammation by blocking the synthesis of COX-1 and COX-2 [98]. Pinus maritime decreased inflammation by enhancing the HO-1 antioxidative system and suppressing the levels of proinflammatory cytokines such as IL-1β and IL-6 [99]. Saururus chinensis, a culinary material, inhibited inflammation by blocking NF-κB activation and the expression of COX-2 and PEG_2_ [100].

### 4.3. Apoptosis Modulators

Chrysin exerts anti-asthmatic effects via several therapeutic pathways, such as immune modulation, anti-inflammation, and apoptosis induction. Further, it induced the apoptosis of airway smooth muscle cells through the dephosphorylation of ERK1/2 [101]. *Codonopsis laceolata* significantly controls ovalbumin-induced inflammation and cell death, such as apoptosis of infiltrated inflammatory cells and pulmonary epithelial cell death [96]. Curcumin induces cell death by decreasing the expression of the anti-apoptotic protein Bcl-2 and induces endoplasmic reticulum stress [102].

The safety of natural products has been well demonstrated, as they have been used for a long time, and it is easy to develop new drugs for asthma treatment using them because of their anti-asthmatic effects. As drug development is based on the pathogenesis of the disease, natural products are classified into three categories by studies on their anti-asthmatic properties: immune modulators, anti-inflammatory effectors, and apoptosis modulators.

## 5. Discussion

According to a WHO Fact Sheet of 2019, 262 million people suffered from asthma and 455 thousand persons died from the disease. Asthma is a severe health problem for children and older adults [4], and as the aging of society intensifies [5], it is likely that the population of patients with asthma will increase. Although allergic asthma inducers involve indoor and outdoor allergens, air pollution and climate change have been intensively considered as additional severe factors that accelerate the prevalence of asthma owing to the rapid industrialization [11,103].

Asthma medications consist of a controller and a reliever [60]. The controller blocks inflammation or modulates the immune reaction [59], and the reliever controls asthma by dilating the pulmonary smooth muscle to expand the small airways [62].

When the allergen invades a bio-organism repeatedly, the immune system is activated, resulting in the activation of immune cells, such as eosinophils and neutrophils, which in turn eliminate the allergens by producing ROS. Although the synthesized ROS may eliminate allergens, they can harm the bio-organism. Finally, repeated immune cell activation causes damage to the pulmonary system, including inflammation and airway remodeling, and can cause chronic asthma [34]. Potent anti-inflammatory drugs, such as corticosteroids, leukotriene modifiers, and mast cell stabilizers, have long been used but have several adverse effects [75,76,77,78].

The levels of Th2 cell-related factors are significantly increased in patients with asthma [22], and Th2 cell-related cytokines, such as IL-4, IL-5, and IL-13, have many functions related to asthma occurrence, such as stimulating the secretion of IgE by activating B cells, inducing eosinophilia and airway hyper-responsiveness, and stimulating goblet cell metaplasia [25,26,27]. Other asthma controllers include immune modulators such as immune suppressors (methotrexate and cyclosporine A) and immune potentiators (glucocorticoids, 1,25-dihydroxy vitamin D3, and toll-like receptor 2/4/9 ligands). However, they have several adverse effects, such as nausea, an increase in serum aminotransferase, diarrhea, and hair problems [59,79]. A combined therapy with a reliever and controller is usually used to increase the anti-asthmatic effect of a drug and decrease its adverse effects [60]. Many studies have been conducted to develop safer and more effective drugs for asthma, including the investigation of natural products with anti-asthmatic effects. However, although natural products have long been used, they have not been confirmed to be safer than chemical drugs. For example, Guryanova et al. [104] reported that natural products, including those from bacterial sources, have dual effects depending on the time of administration. If they are used before the onset of asthma, which can be referred to as sensitization in an experimental model system, the severity of asthma can be significantly reduced; however, if used after the disease is established, they increase asthma severity. Therefore, the timing of administering natural products, including those from bacterial sources, to treat asthma is one of the most important factors to consider.

To develop asthma drugs from natural products, we should restrict the range of candidates that have long been used but lack reports on toxicity. Consequently, their toxicity should be evaluated before approval for market entry. In conclusion, natural products have many biological effects and can be used as drugs for asthma to exert immune modulation, anti-inflammation, and apoptosis modulation. Therefore, the development of drugs from natural products for the treatment of asthma may be a useful strategy.

## Figures and Tables

**Figure 1 ijms-24-12469-f001:**
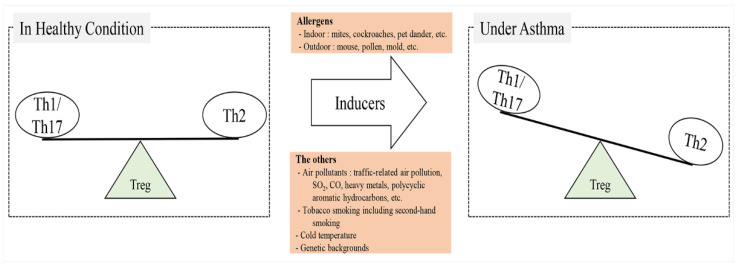
Asthma occurrence. In healthy conditions, the balance of Th1, Th17, and Th2 cells can be maintained, but asthma inducers stimulate their imbalance. Treg cells modulate the immune balance between Th1, Th17, and Th2 cells, as they can regulate the development of naïve T cells to Th1, Th17, or Th2 cells. Th1, helper 1 T cell; Th2, helper 2 T cell; Th17, helper 17 T cell; Treg, regulatory T cell.

**Figure 2 ijms-24-12469-f002:**
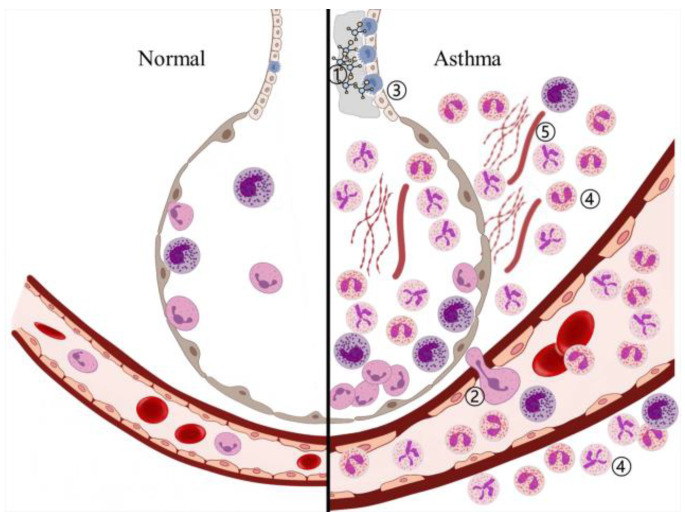
Bronchioalveolar inflammation in asthma. ➀ Invasion by inflammation inducers, ➁ release of immune cells such as neutrophils and eosinophils from blood vessels, ➂ hypersecretion of mucus from goblet cells, ➃ infiltration of immune cells near bronchioalveolar ducts and blood vessels, and ➄ airway smooth muscle contraction. 
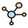
, inducer; 
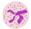
, neutrophil; 
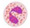
, eosinophil; 
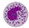
, basophil; 
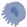
, Goblet cell.

**Figure 3 ijms-24-12469-f003:**
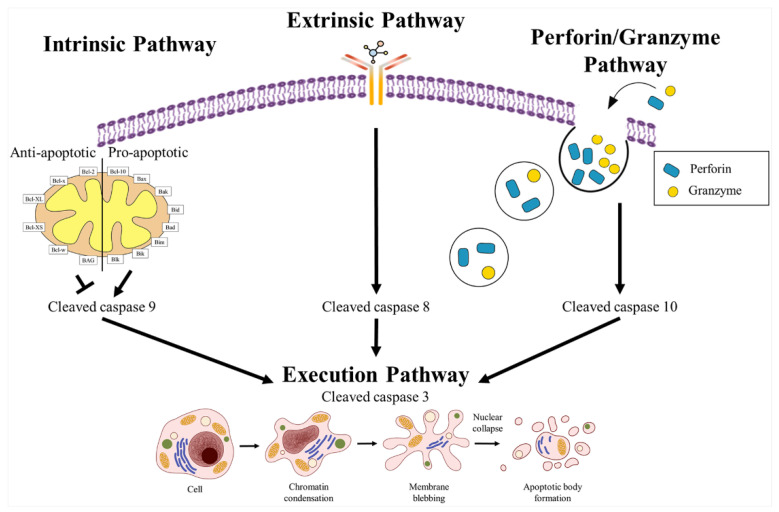
Mechanisms of apoptosis. The intrinsic pathway is related to mitochondrial activity, the extrinsic pathway is initiated by the binding of an apoptotic ligand to the apoptotic receptor, and the perforin/granzyme pathway is related to perforin and granzyme. Cell death (apoptosis) is completed through the execution pathway, which leads to chromatin condensation, membrane blebbing, and nuclear collapse. 
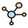
, apoptotic ligand; 
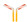
, apoptotic receptor; 
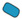
, perforin; 
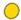
, granzyme.

**Table 1 ijms-24-12469-t001:** Pulmonary system inflammation according to onset time. According to the onset time of inflammation in the respiratory system, asthma can be classified into two stages, acute and chronic. The acute stage consists of the early phase and the late phase. The data in this table are from Reference [34].

Stage	Acute		Chronic
Phase	Early	Late	
Onset Time after Allergens Contact	Several minutes	Onset time: 2~6 h Peak time: 6~9 h	A few months to years
Physiological Symptoms	Vasodilation, vascular permeability, bronchoconstriction, mucous hypersecretion, etc.	Wheeze, shortness of breath, cough	Wheeze, shortness of breath, cough, mucus hypersecretion, sleep apnea to death
Causes of the Symptoms	Cytokine and chemokine release after the cross linkage of IgE–mast cell/basophil–allergens	Rise of Th2-related cytokines and increment of WBCs	Alteration of the extracellular matrix and of the constitutive cells of the affected organ

**Table 3 ijms-24-12469-t003:** Anti-asthmatic effects of natural products based on the therapeutic pathway. The source indicates the origin of the applied material. The applied type is the treatment form described in this study: if the applied form is the isolated compound, it is described as “Isolated compound/active substance”. However, if the active substance is known but is different from the applied form, they are indicated as “applied type and active substance”. P.O., oral administration; Refs., references.

Classification		Source	Applied Type/ Active Substance	Study	Dose (Max) & Route (mg/kg)	Mode of Action	Refs.
Immune modulators	Chrysin	Propolis, passionflower, etc.	Isolated compound/ chrysin	Animal	100, P.O.	Decreasing the levels of IgE, IL-4, and IL-13	[80]
	1′-acetoxychavicol acetate	*Alpinia galanga*	Isolated compound/ 1′-acetoxychavicol acetate	Animal	50, P.O.	Decreasing the levels of Th1 cell-related cytokines such as IL-12 and IFN-γ and of Th2-related cytokines such as IL-4, IL-5, and IL-13	[81]
	Alginate oligosaccharide	Fermented alginate	Fermented form	Animal	400, P.O.	Inhibiting the releases of IL-15 and IL-13	[82]
	*Allium cepa* L. and quercetin	*Allium cepa* L. and quercetin	Methanol extract and isolated compound/quercetin	Cell & Animal	1000 + 15 μg/mL, 1000 + 30 mg/kg, P.O.	Decreasing the levels of IL-4 and IL-13	[83]
	*Allium hookeri*	*Allium hookeri*	Ethanol extract	Animal	300, P.O.	Inhibiting the releases of IL-4, IL-5, and IL-13	[84]
	*Anoectochilus formosanus* HAYATA	*Anoectochilus formosanus* HAYATA	Water extract	Animal	1000, P.O.	Decreasing the levels of IgE, IL-4, and IL-5	[85]
	*Caenorhabditis elegans*	*Caenorhabditis elegans*	Crude extract	Animal	50 μg/head	Decreasing the levels of IgE, IL-5, and IL-13 but increasing the levels of IFN-γ	[86]
	*Camellia japonica*	*Camellia japonica*	Extract oil and isolated compound/oleic acid	Animal	500, P.O.	Decreasing the levels of IL-4 via GATA-3 inactivation	[87]
	*Citrus tachibana*	*Citrus tachibana*	Ethanol extract narirutin, hesperidin	Animal	400, P.O.	Decreasing the levels of IL-4 and IL-5 but increasing the levels of IL-12 and IFN-γ	[88]
	Curcumin	*Curcuma longa*	Isolated compound/diferuloylmethane	Animal	2000, P.O.	Inactivating Th2 cell transcription factor, GATA-3	[89]
	*Erythronium japonicum*	*Erythronium japonicum*	Ethanol extract chlorogenic acid	Animal	600, P.O.	Decreasing the levels of IL-12p35, IFN-γ, IL-4, IL-5, and IL-13	[90]
	Korean red ginseng	*Panax ginseng*	Water extract Active substances: Rb1, Rg1	Animal	50, P.O.	Decreasing the levels of IL-12, IL-4, and IL-6	[91]
	*Mycoleptodonoides aitchisonii*	*Mycoleptodonoides aitchisonii*	Water extract niacin, oleic acid, linoleic acid	Animal	1000, P.O.	Decreasing the levels of IL-4, IL-5, and IL-13	[92]
	*Opuntia humifusa*	*Opuntia humifusa*	Ethanol extract rutin, quercetin	Animal	500, P.O.	Decreasing the levels of IL-12, IFN-γ, IL-4, IL-13, IL-6, and TNF-α	[93]
Anti-inflammatory effectors	Chrysin	Propolis, passionflower, etc.	Isolated compound/chrysin	Animal	100, P.O.	Blocking inflammatory cell infiltration	[80]
	3-methoxy-catalposide	*Psueolysimachion rotundum* var. subintegrum	Isolated compound/3-methoxy-catalposide	Cell	20 μM	Decreasing the expression of COX-2 and iNOS and downregulating the levels of IL-1β, IL-6, and TNF-α	[94]
	*Aster yomensa* (Kitam.) Honda	*Aster yomensa* (Kitam.) Honda	Ethanol extract	Cell	300 ng/mL	Decreasing the levels of NO and IL-1β	[95]
	*Codonopsis laceolata*	*Codonopsis laceolata*	Water extract Active substance: lobetyolin	Animal	300, P.O.	Blocking the NF-κB/COX-2 and PGE_2_ pathway	[96]
	*Echinodorus scaber* Rataj	*Echinodorus scaber* Rataj	Ethanol extract	Animal	30, P.O.	Suppressing inflammatory cells’ migration	[97]
	Korean red ginseng	*Panax ginseng*	Water extract Active substances: Rb1, Rg1	Animal	50, P.O.	Blocking the NF-κB/COX-2 and PGE_2_ pathway	[91]
	*Pericampylus glaucus*	*Pericampylus glaucus*	Hexane, chloroform or ethanol extract	Cell	250 μg/mL	Blocking the synthesis of COX-1 and COX-2	[98]
	*Pinus maritime*	*Pinus maritime*	Water extract	Cell	100 pg/mL	Enhancing the HO-1 antioxidative system and decreasing the levels of IL-1β and IL-6	[99]
	*Saururus chinenesis*	*Saururus chinenesis*	Water extract rutin, quercitrin, quercetin	Animal	300, P.O.	Blocking the NF-κB/COX-2 and PGE_2_ pathway	[100]
Apoptosis modulators	Chrysin	Propolis, passionflower, etc.	Isolated compound/chrysin	Animal/Cell	100, P.O./40 μM	Inducing the apoptosis of airway smooth muscle cells	[80,101]
	*Codonopsis laceolata*	*Codonopsis laceolata*	Water extract Active substance: lobetyolin	Animal	300, P.O.	Inducing infiltrated inflammatory cells’ apoptosis and pulmonary epithelial cells’ death	[96]
	Curcumin	*Curcuma longa*	Isolated compound/diferuloylmethane	Cell	20 μM	Decreasing the expression of the anti-apoptotic protein Bcl-2 and inducing endoplasmic reticulum stress	[102]

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
