# Peer review of "Drug Development from Natural Products Based on the Pathogenic Mechanism of Asthma"

_ijms, 2023, doi:10.3390/ijms241512469_

Round 1
Reviewer 1 Report
The article is dedicated to asthma drug development from the natural products. The topic discussed is crucial for the treatment and prevention of asthma.
I would like to make a few comments:
1) Information on the impact of genetic backgrounds on the pathogenesis and treatment of asthma is incomplete. More then 100 gene polymorphisms are associated with asthma according latest research. Discuss the article please:
Shi F, Zhang Y, Qiu C. Gene polymorphisms in asthma: a narrative review. Ann Transl Med. 2022 Jun;10(12):711. doi: 10.21037/atm-22-2170.
2) When you are writing about mechanisms of asthma pathogenesis (lines 99-126) please discuss that asthma is reflected by distinct phenotypes orchestrated by Th2 or non-Th2 (Th1 and Th17) immune responses. Read and cite the article:
Luo W, Hu J, Xu W and Dong J (2022) Distinct spatial and temporal roles for Th1, Th2, and Th17 cells in asthma. Front. Immunol. 13:974066. doi: 10.3389/fimmu.2022.974066
3) When you discuss inflammation, the influence of ligands of innate immunity receptors has a significant impact on the formation of the immune system. In particular, it is necessary to discuss the influence of innate immunity receptors TLR and NLR on the pathogenesis of asthma and cite articles:
Clarke TB. Early innate immunity to bacterial infection in the lung is regulated systemically by the commensal microbiota via nod-like receptor ligands. Infect Immun. 2014 Nov;82(11):4596-606. doi: 10.1128/IAI.02212-14
Mu C, Yang Y, Zhu W. Crosstalk Between The Immune Receptors and Gut Microbiota. Curr Protein Pept Sci. 2015;16(7):622-31. doi: 10.2174/1389203716666150630134356
It is very important, that NLR ligands from the gastrointestinal, but not upper respiratory, tract rescued host defenses in the lung.
4) Discuss the complex nature of innate immune receptor regulation of allergic inflammation. In particular, prolonged stimulation with low doses of innate immunity receptors agonists before sensitization by an allergen reduces the severity of the allergic process. In the case of innate immunity receptors stimulation together with the action of an allergen, allergic inflammation increases in asthma model.
Need to discuss, cite the article:
Guryanova, S.V.; Gigani, O.B.; Gudima, G.O.; Kataeva, A.M.; Kolesnikova, N.V. Dual Effect of Low Molecular Weight Bioregulators of Bacterial Origin in Experimental Model of Asthma. Life 2022, 12, 192. https://doi.org/10.3390/life12020192
5) Please also cover methods of prevention of asthma, including dietary advice.
Author Response
The article is dedicated to asthma drug development from the natural products. The topic discussed is crucial for the treatment and prevention of asthma.
I would like to make a few comments:
1) Information on the impact of genetic backgrounds on the pathogenesis and treatment of asthma is incomplete. More then 100 gene polymorphisms are associated with asthma according latest research. Discuss the article please:
Shi F, Zhang Y, Qiu C. Gene polymorphisms in asthma: a narrative review. Ann Transl Med. 2022 Jun;10(12):711. doi: 10.21037/atm-22-2170.
Ans) Thank you so much for informative and generous comment and I amended the part.
2) When you are writing about mechanisms of asthma pathogenesis (lines 99-126) please discuss that asthma is reflected by distinct phenotypes orchestrated by Th2 or non-Th2 (Th1 and Th17) immune responses. Read and cite the article:
Luo W, Hu J, Xu W and Dong J (2022) Distinct spatial and temporal roles for Th1, Th2, and Th17 cells in asthma. Front. Immunol. 13:974066. doi: 10.3389/fimmu.2022.974066
Ans) Thank you so much for the informative comment and I amended the manuscript.
3) When you discuss inflammation, the influence of ligands of innate immunity receptors has a significant impact on the formation of the immune system. In particular, it is necessary to discuss the influence of innate immunity receptors TLR and NLR on the pathogenesis of asthma and cite articles:
Clarke TB. Early innate immunity to bacterial infection in the lung is regulated systemically by the commensal microbiota via nod-like receptor ligands. Infect Immun. 2014 Nov;82(11):4596-606. doi: 10.1128/IAI.02212-14
Mu C, Yang Y, Zhu W. Crosstalk Between The Immune Receptors and Gut Microbiota. Curr Protein Pept Sci. 2015;16(7):622-31. doi: 10.2174/1389203716666150630134356
It is very important, that NLR ligands from the gastrointestinal, but not upper respiratory, tract rescued host defenses in the lung.
Ans) Thank you so much for the informative comment and I amended the manuscript.
4) Discuss the complex nature of innate immune receptor regulation of allergic inflammation. In particular, prolonged stimulation with low doses of innate immunity receptors agonists before sensitization by an allergen reduces the severity of the allergic process. In the case of innate immunity receptors stimulation together with the action of an allergen, allergic inflammation increases in asthma model.
Need to discuss, cite the article:
Guryanova, S.V.; Gigani, O.B.; Gudima, G.O.; Kataeva, A.M.; Kolesnikova, N.V. Dual Effect of Low Molecular Weight Bioregulators of Bacterial Origin in Experimental Model of Asthma. Life 2022, 12, 192. https://doi.org/10.3390/life12020192
Ans) Thank you so much for the informative comment and I described the dual effect of natural products on downregulating asthma severity in the discussion section.
5) Please also cover methods of prevention of asthma, including dietary advice.
Ans) Thank you so much for the informative comment and I described the dual effect of natural products on downregulating asthma severity in the discussion section.

Reviewer 2 Report
The article „Asthma Drug Development from the Natural Products” by Kim et al. has a very interesting title and quite promising abstract. Asthma is a global problem and new drugs are needed. One of their sources could be traditional medicine based on the old knowledge of our ancestors.
Unfortunately, the article needs corrections before publication, and I would have some remarks to the Authors. At this moment the article is somehow chaotic and difficult to understand.
First of all, English language needs extensive corrections. Try to write clear, short and grammatically correct sentences to formulate your ideas and avoid to duplicate information or present those irrelevant to the main topic of the paper.
Second, have in mind the main topic of the paper (natural products in asthma treatment) and in this context, try to present the pathophysiology of asthma without giving irrelevant or too detailed information. Consider to just add a or some review papers where additional information about asthma pathology is explained.
Let me discuss the parts of the paper.
Introduction.
- Divide this incredible long sentence into some shorter.
Pathogenesis of asthma.
- What about non-allergic asthma or neutrophilic asthma?
- Maybe it would be an idea to combine the Th1/Th2 imbalance with airway inflammation? I am not sure if the problem of apoptosis is the most important one is asthma but if you decided to discuss it, a classification of Bcl-2 proteins is not really needed as you do not discuss it any further.
- What about mucus hypersecretion and redox imbalance(pro-/anti-oxidant)? Natural anti-asthmatic products are known to reduce both.
- You made excellent figures, only the legends should be much shorter and not repeat the information from the text.
Asthma medication.
- Once again, short sentences, only relevant information, avoid strange sentences like in line 262, and do not concentrate only on side effects of the substances. Try to be objective and write some positive sides of their use.
Natural drugs for asthma treatment.
- Write in a short introduction why did you chose for discussion these substances and do not forget to put Latin names in italics. It makes it easier to read. Please write more about the respective substance or plant, of course, in the context of asthma treatment and its aspects. Give more information than simply enumating it, maybe about its traditional use?
- In Table 3, a column with the name of the active substance or class of substances (if known) should be added. Is C. elegans really relevant here? Is it mentioned in the text?
Discussion.
Unfortunately, this part of the paper is quite chaotic. Was the main idea of the text the opinion that natural substances, including these enumerated ones, better for asthma treatment because of lower side effects? Indeed, it is not always the fact but it is a common belief as herbal extracts, which are often considered as “natural products”, are used for a long long time as traditional medicine. Please explain what are “chemical synthetic drugs”. I am not really sure if a full enumeration of side effects is needed here, especially that you mentioned most of them above. In the discussion the main ideas of the paper should be discussed. Consider to discuss the substances you mentioned in the context of their use in the treatment of asthma or, in general, natural products in asthma treatment. Consider to mention also other reasons than only lower side effects.
To sum-up, the article needs to be corrected and re-written prior to publication.
English language needs extensive corrections. Sentences need to be clear, short and easy to understand. At the moment, their are not.
Author Response
The article „Asthma Drug Development from the Natural Products” by Kim et al. has a very interesting title and quite promising abstract. Asthma is a global problem and new drugs are needed. One of their sources could be traditional medicine based on the old knowledge of our ancestors.
Unfortunately, the article needs corrections before publication, and I would have some remarks to the Authors. At this moment the article is somehow chaotic and difficult to understand.
First of all, English language needs extensive corrections. Try to write clear, short and grammatically correct sentences to formulate your ideas and avoid to duplicate information or present those irrelevant to the main topic of the paper.
Ans) Before resubmitting the manuscript it had been edited by editing company. Please check-up the certificate of editing by company.
Second, have in mind the main topic of the paper (natural products in asthma treatment) and in this context, try to present the pathophysiology of asthma without giving irrelevant or too detailed information. Consider to just add a or some review papers where additional information about asthma pathology is explained.
Ans) Thank you so much and I changed the main topic, “Drug Development from the Natural Products based on Pathogenic Mechanism of Asthma”.
Let me discuss the parts of the paper.
Introduction.
- Divide this incredible long sentence into some shorter.
Ans) Thank you so much and we changed the paragraphs to be shorter.
Pathogenesis of asthma.
- What about non-allergic asthma or neutrophilic asthma?
Ans) Thank you so much for informative comment and we amended the manuscript.
- Maybe it would be an idea to combine the Th1/Th2 imbalance with airway inflammation? I am not sure if the problem of apoptosis is the most important one is asthma but if you decided to discuss it, a classification of Bcl-2 proteins is not really needed as you do not discuss it any further.
Ans) Thank you so much for generous and informative comments. In this section, ‘Mechanisms of Asthma Pathogenesis’ we tried to discuss on three pathogenic factors for asthma such as abnormality of immune response like imbalance of Th1, Th2, and Th17, chronic respiratory inflammation which is caused by various factors such as cytokines and chemokines, and hyperplasia of respiratory epithelial cells (anti-apoptosis). As you know the intrinsic pathway in apoptosis is related with Bcl-2 family and some reports have proved the anti-asthmatic effect which is related with Bcl-2 family including our research results which is not published yet. We described this section to explain the relation of anti-asthmatic effect of natural products and anti-apoptotic mechanism through modulation of Bcl-2 family.
- What about mucus hypersecretion and redox imbalance(pro-/anti-oxidant)? Natural anti-asthmatic products are known to reduce both.
Ans) Thank you so much and we amended the manuscript.
- You made excellent figures, only the legends should be much shorter and not repeat the information from the text.
Ans) Thank you so much and we amended the legends of Tables shorter.
Asthma medication.
- Once again, short sentences, only relevant information, avoid strange sentences like in line 262, and do not concentrate only on side effects of the substances. Try to be objective and write some positive sides of their use.
Ans) Thank you so much and we amended the manuscript.
Natural drugs for asthma treatment.
- Write in a short introduction why did you chose for discussion these substances and do not forget to put Latin names in italics. It makes it easier to read. Please write more about the respective substance or plant, of course, in the context of asthma treatment and its aspects. Give more information than simply enumating it, maybe about its traditional use?
Ans) Thank you so much for the comments and we amended the manuscript.
- In Table 3, a column with the name of the active substance or class of substances (if known) should be added. Is C. elegans really relevant here? Is it mentioned in the text?
Ans) Thank you so much for generous comment and we amended the manuscript. Although C. elegans is not used for anti-asthmatic material as anti-asthmatic effect of that had been reported we cited that in this chapter.
Discussion.
Unfortunately, this part of the paper is quite chaotic. Was the main idea of the text the opinion that natural substances, including these enumerated ones, better for asthma treatment because of lower side effects? Indeed, it is not always the fact but it is a common belief as herbal extracts, which are often considered as “natural products”, are used for a long long time as traditional medicine. Please explain what are “chemical synthetic drugs”. I am not really sure if a full enumeration of side effects is needed here, especially that you mentioned most of them above. In the discussion the main ideas of the paper should be discussed. Consider to discuss the substances you mentioned in the context of their use in the treatment of asthma or, in general, natural products in asthma treatment. Consider to mention also other reasons than only lower side effects.
Ans) Thank you so much for informative comments and we amended the manuscript.
To sum-up, the article needs to be corrected and re-written prior to publication.
Ans) Before resubmitting the manuscript it had been edited by editing company. Please check-up the certificate of editing by company.

Round 2
Reviewer 2 Report
The authors of the paper “Drug Development from Natural Products Based on the Pathogenic Mechanism of Asthma” made a great effort to correct their article and to explain their ideas in a much clearer way. The text is much better now, it is easier to read and to understand. The first part about the pathogenesis of asthma gives a good background on the subject and on the treatment options. In the second part, the potential of selected natural substances towards the treatment of asthma or its symptoms is briefly presented. A good table summarizes the description leading altogether to the statement that natural substances can be a good treatment option in asthma as their have fewer side effects that modern drugs have. At this point, consider to change in line 1108 the word “these” to “the following" or something similar as “these” in this context refers to asthma relievers which are only rarely immunomodulators or anti-inflammatory agents.